# Improved Wood-Bond Strengths Using Soy and Canola Flours with pMDI and PAE

**DOI:** 10.3390/polym14071272

**Published:** 2022-03-22

**Authors:** Mahsa Barzegar, Linda F. Lorenz, Rabi Behrooz, Charles R. Frihart

**Affiliations:** 1Department of Wood and Paper Science and Technology, Faculty of Natural Resources and Marine Sciences, Tarbiat Modares University, Imam Reza Blvd, Noor 46414-356, Iran; mahsa.barzegar85@gmail.com (M.B.); rabi.behrooz@modares.ac.ir (R.B.); 2Forest Products Laboratory, USDA Forest Service, One Gifford Pinchot Drive, Madison, WI 53726-2398, USA; charles.r.frihart@usda.gov

**Keywords:** soy flour, canola flour, wood adhesive, shear strength, water resistance, pMDI, PAE

## Abstract

The surprising lack of literature on using the very common wood adhesive polymeric methylenediphenyl diisocyanate (pMDI) with protein adhesives may be because of perceived poor improvement of protein wet strength. Reacting pMDI with the flour (soy or canola) before adding water unexpectedly improves wood bonding compared to adding the pMDI to an aqueous protein slurry. Mixing the liquid pMDI with the oilseed flour produces a free-flowing powder with up to 50% of pMDI to flour by weight. The mixture slowly reacts since the isocyanate band in the infrared spectra remains for several days but diminishes with time. Adding pMDI increases the dry and wet strength of wood bonds using Automated Bonding Evaluation System (ABES) testing and levels off at about 50%. Similarly, adding the polyamidoamine-epichlorohydrin (PAE) cross-linker to the oilseed flour increases dry and wet bond strength, but the effect levels off at about 20% of PAE. However, the combination of these two cross-linkers added to the flours results in greater dry and wet shear strength than either one alone. In addition to tests using ABES (ASTM D 7998), the increase in strengths is also observed—but with a diminished effect—in bonding plywood using the interior plywood strength test ASTM D 906.

## 1. Introduction

Although protein adhesives (soy, casein, blood, collagen, etc.) have been replaced by the invention and extensive development of synthetic adhesives from fossil fuels, concerns about formaldehyde emissions from urea-formaldehyde, the largest-volume wood adhesive, have renewed interest in bio-based adhesives [1,2,3]. While most of the research has been on soybean-based adhesives, other oilseeds (canola, cottonseed, etc.) have been studied [4,5,6]. Unfortunately these natural materials, except for egg whites [7], do not provide high-strength wood bonds under wet test conditions without further modification. In recent years, many ways to improve the bond strength of soy-based adhesives have been published [1,5], including adding chaotropic agents (urea, guanidine hydrochloride, dicyandiamide, etc.), changing the amount and type of salts (Hofmeister series), adding surfactants (including sodium dodecyl sulfate and sodium dodecyl benzene sulfonate), modifying the proteins (dopamine, cysteamine, CaCO_3_, MgO and POCl_3_), breaking apart protein agglomerates (sodium metabisulfite, acid, base hydrolysis, or enzymatic cleavage), grafting reactive side chains (epoxies, 2-octen-1-ylsuccinic anhydride, 3-aminopropyltriethoxysilane, 1-ethyl-3-[3-dimethylaminopropyl] carbodiimide hydrochloride, maleic anhydride, etc.), or crosslinking the proteins/carbohydrates (formaldehyde, glyoxal, polyamidoamine-epichlorohydrin (PAE) resin, etc.) [1,8]. According to the literature, most of these modifications provide better wood-bond strengths than the unmodified soy adhesives. The soy adhesives in the literature can range all the way from the native soy flour to the highly modified commercial soy protein isolate as the base soy. However, we find it hard to know the accuracy of these claims since they are often singular papers with few controls, use different wood species and test conditions, and usually do not report on viscosity changes, which are very important for most wood-adhesive tests. In this paper and our many other papers, we have used the ABES test, ASTM D 7998-19, “Standard Test Method for Measuring the Effect of Temperature on the Cohesive Strength Development of Adhesives using Lap Shear Bonds under Tensile Loading,” because we have obtained repeatable results with this test, which is less sensitive to adhesive viscosity and wetting phenomena, has a smooth surface for uniform wood veneer, and can test a wide variety of bonding temperatures and times [9,10]. The thin samples can be soaked in water for water-durability determination. A concern about some of the literature results is that we come to different conclusions since some of the modifications do not provide enhanced moisture durability using ASTM D-7998 and plywood testing [11,12]. 

Although we have studied many methods for improving soy adhesives, our recent interest has been with polymethylenediphenyl diisocyanate (pMDI) and why such a common wood adhesive has not been studied more by researchers for improving protein adhesives [13]. One report involved adding small amounts of diisocyanates to soy adhesives for promoting wood adhesion strength [14] and others examined how much soy protein could be added to pMDI adhesive before it interfered with the pMDI adhesive strength [15,16]. 

In our prior research, the well-studied soy flour was compared with the rarely studied canola (a type of rapeseed) flour, incorporating into the adhesive not only the commercially used PAE resin but also pMDI [17]. We found that reacting the pMDI with the dry soy or canola flour first before adding the water produced stronger bonds than adding an equivalent amount of pMDI to the soy or canola flour aqueous dispersion. This research has indicated that the positive effect of adding the pMDI or PAE leveled off as more was added, but the effect needed to be studied further, since the emphasis in our previous paper was on comparing the rarely studied canola to the widely studied soy. Therefore, we explored these areas in more detail, and also determined if adding both cross-linkers was better than each one separately. 

## 2. Materials and Methods

### 2.1. Materials

Defatted canola flour (*Brassica napus* L.) was obtained from Behpak Industrial Co (Behshahr, Mazandaran, Iran), soy flour (*Glycine max*) Prolia™ 200/90 from Cargill (Cedar Rapids, IA, USA), and soy protein isolate, PRO-FAM^®^ 974 from ADM Inc. (Decatur, IL, USA). Hard maple (*Acer saccharum Marsh.*) rotary-cut veneers of 0.6 mm thickness for the ABES tests and yellow poplar (*Liriodendron tulipifera* L.) rotary-cut veneers 3 mm thick for the plywood tests were provided by Columbia Forest Products (Old Fort, NC, USA). Sodium hydroxide was purchased from Sigma Aldrich (St. Louis, MO, USA). The polymethylenediphenyl diisocyanate (pMDI) was provided by Huntsman (The Woodlands, TX, USA), and polyamidoamine-epichlorohydrin (PAE, CA1920) was obtained from Solenis (Wilmington, DE, USA).

### 2.2. Preparation of Soy and Canola Flour Adhesives 

The solids content of the adhesives was 25% based on the dry weight of soy or canola flour. Soy or canola was reacted with pMDI at 5% to 50% of the weight of the flour for about 30 min. The mixture was then dispersed in water at 25% based on the dry weight of the flour and the mixture was stirred for 5 min. After the PAE was added at 5% of the dry weight of the flour, the mixture was stirred for an additional 5 min, and the pH was adjusted to 6.5 with 5 M sodium hydroxide. The total adhesive solids content was 26.8% to 35.8% and the viscosity was about 100,000 cPs for the highest amount of pMDI added and somewhat less for the lesser amounts of pMDI added.

### 2.3. Evaluation of Shear Strength of Synthesized Adhesives 

#### 2.3.1. ABES

Maple veneer specimens with dimensions of 117 × 20 mm^2^ were cut using a die cutter from veneer sheets conditioned at 21 °C and 50% RH. Adhesive was applied to 5 mm at the end of one specimen with a resin spread rate of ~100 g/m^2^, which was overlapped 5 mm with the end of another veneer specimen. The assembled sample was hot pressed in the ABES (Adhesive Evaluation Systems, Inc., Corvallis, OR, USA) at 120 °C for 2 min [9,10].

#### 2.3.2. Plywood 

Poplar 3-ply plywood panels (3 mm/veneer ply) were bonded with soy or canola flour at 25% flour solids plus 50% pMDI and 5% PAE (based on the flour solids) for a total adhesive solids content of 35.8% and a viscosity of about 100,000 cPs. The spread rate was 215 g/m^2^. The panels were bonded at 120 °C for 5 min at 0.86 MPa (125 psi) after a closed assembly time of 15 min. The panels were conditioned at 26.7 °C and 30% RH for one week before being cut into samples. For testing shear strength, the panels were cut into 8.26 × 2.54 cm samples according to ASTM D 906-20 [18], and half were tested dry and half tested wet after soaking in room temperature water for 24 h. In a modification of D 906, all samples were pulled with the lathe checks open to obtain consistent data.

### 2.4. ATR-FTIR

Soy or canola flour was reacted with 50% pMDI and the dry mixture was analyzed by ATR-FTIR (Perkin Elmer Spectrum two) with the wave number range of 450–4000 cm^−1^ and a 4 cm^−1^ resolution with 16 scans. 

## 3. Results and Discussion

To understand protein adhesion performance, it is important to understand the structures of defatted oilseed flours. They are mainly composed of high-molecular-weight proteins and carbohydrates as reservoirs to provide low-molecular-weight monomers to help initiate seed growth. For bonding, the smaller molecules may serve as surfactants and are able to soak into the wood, while the adhesive strength is provided by the “coalescence” of the high-molecular-weight polymers. However, this is probably not true coalescence in fusing the particles, but more likely weak coalescence by close association of the colloid particles because of the observed water sensitivity when testing bonded samples. Like polyvinyl acetate (PVA) adhesives, the soy adhesives provide enough dry strength to cause wood failure, but the adhesives are plasticized under wet conditions and result in almost no wood failure. The individual protein and carbohydrate molecules are firmly held together by polar and hydrophobic bonds. Nonetheless, lacking actual fusion of the molecules or sufficient covalent bonds, water can pry apart these condensed structures. Just as with PVA, some type of cross-linking is needed to form water-resistant bonds. In addition, with higher bonding temperatures or longer bonding times, the soy adhesives form stronger bonds most likely due to fusing (interdiffusion) of these individual molecules [11].

### 3.1. Results of Evaluation of ABES Shear Strength

Our prior paper [17] reported on understanding how canola flour adhesive compared with the widely studied soy flour, since canola is much more widely grown in Canada, Europe, and parts of Asia than soy. As shown in Table 1, the canola flour has good dry strength but poor wet strength, similar to soy flour. The wet strength is far below what is obtained with the commercial soy protein isolate (CSPI) and a minimum of 3.0 MPa for pursuing other performance testing. 

#### 3.1.1. Reaction of Soy and Canola Flour with pMDI

Like the soy flour, the canola flour needs some modification to obtain good wet-strength values. As noted in the introduction (Section 1), of all the possible cross-linking chemicals for use with soy, the diisocyanates, especially pMDI, have only sporadically been studied. Our initial work showed that adding pMDI to the aqueous adhesive did not result in wood bonds as good as adding it to the flour first [17]. An explanation of this is that the pMDI probably reacts with the abundant water before it has a chance to infiltrate the protein structure. This reduces the amount of pMDI available for cross-linking of the protein. Although the self-reaction of pMDI promoted by water is necessary for producing strong wood bonds, sufficient pMDI should be available to cross-link the soy protein as much as possible before the water is added. 

Knowing that pMDI can be mixed with protein flours to form an adhesive by adding water just prior to application was an important development. Two obvious questions were how much pMDI can be added to the flours while still having a free-flowing powder, which is much easier to use than a sticky mass which we initially expected, and how stable these mixtures are over time. Increasing the ratio of pMDI to soy or canola answers these two questions at the same time. Adding 50 g of pMDI to 100 g of flour still produced a free-flowing product. However, this material when dispersed in water had a much higher viscosity and was difficult to spread on the wood compared to when the amount of pMDI was 25 g to 100 g of flour, establishing 25% as the practical limit. In addition, as the amount of pMDI increased, it was observed that the ABES bond shear strength under dry and wet conditions increased as shown in Table 2, and at the highest amount exceeded our preferred minimum wet strength for further testing.

The next question was how stable these pMDI–soy flour mixtures are over time. In Figure 1, dry and wet ABES shear strengths of the 50% pMDI–soy and canola adhesives show a general decrease with time, and unexpectedly, the canola strengths were higher than the soy strengths starting one day after mixing. This suggests that the pMDI can take time to be fully absorbed into the soy and canola structure. Why the canola outperforms the soy is not clear, but the canola protein may be more accessible for reaction with the pMDI than is the soy protein, which is known to be unreactive to most denaturing agents because it is not very accessible [2].

Because we were considering this as a potential commercial process, no special treatment was conducted to remove all the moisture, although vacuum drying the soy flour and then adding pMDI did not result in increased ABES strengths. The loss of strength with time makes it unlikely that this could be more than an in-plant process, but it would only require a dry blender and some storage residence time before making it into an aqueous adhesive (U.S. patent application number 17197956 was filed on 15 June 2021).

To understand the effect of the pMDI, it is important to delve into the structure of the soy flour [19,20]. The individual protein chains and their agglomeration predominately into the glycinin and conglycinin components has been widely reported. However, the protein–protein flocculation into larger protein structures and protein–carbohydrate conjugates results in much larger molecules, which lead to the shear-thinning characteristics of the soy dispersions as the flocculants are broken apart. Thus, the flour–pMDI interaction probably involves the pMDI not infiltrating the very compact individual proteins, which have little free volume, or even the compact glycinin and conglycinin structures, but mainly the larger flocculates or conjugates. The pMDI can then be absorbed in these loosely bonded structures and tie them together to increase the strength by firmly bonding the components by covalent bonds into larger molecules. This is likely to be the mode of strengthening whether the cross-linker is pMDI or PAE.

The effect of the pMDI is probably limited because of the inability to find a reactive group on another polymer chain after attaching to the first polymer chain before the pMDI reacts with water. Asafu-Adjaye et al. [15] have shown that higher amounts of soy disrupt the ability of pMDI to form its normally strong linkage between wood surfaces. Thus, the effect is limited when the pMDI can no longer effectively tie the flour components together before a strong pMDI matrix takes over.

#### 3.1.2. Reaction of Soy Flour with PAE

The effectiveness of PAE to enhance the strength of soy flour adhesives has been reported in the literature and is demonstrated by its commercial use [12]. In contrast to pMDI, PAE is stable in water unless subjected to high heat or high pH, both of which lead to self-polymerization. The effect of the PAE to canola or soy flour ratio on the bond wet shear strength was reported in a prior paper [17]. An optimum amount of PAE added to the flours was determined with the ABES test to be 15 wt% of soy weight (3.5 MPa wet shear strength) and 20 wt% of canola weight (2.0 MPa wet shear strength).

In Figure 2, the data show that very low amounts of PAE improve the bond strength under wet test conditions. The ABES tests are mainly a cohesive strength measurement for the adhesives, and the PAE is remarkably effective in tying the soy molecules together. The PAE increases the wet strength of paper towels and by itself is a good wood bonding adhesive as shown by ABES dry and wet shear strengths of 3.6 and 2.1 MPa.

#### 3.1.3. Reaction of Soy Flour and Canola with pMDI and PAE

The data in Section 3.1.1 and Section 3.1.2 shows that both the pMDI and PAE are separately good cross-linkers with the two flours, especially when tested under dry test conditions, but they are also very effective at improving strengths under wet test conditions. We explored the effect of combining the two cross-linkers with both flours by adding the PAE and water to the dry-blended flour–pMDI. As shown in Table 3, the two cross-linkers indeed provided strengths higher than with either cross-linker alone; this indicates that the PAE and pMDI probably react with different functional groups on the proteins and carbohydrates.

### 3.2. Plywood Adhesive Tests

Although the ABES test is quite informative, it is not an absolute predictor of plywood bond strength. Because of the smooth wood surface, even bonding pressure, and good heat transfer, the soy and canola tests with the ABES emphasized the cohesive strength of the adhesive as a thin film. On the other hand, plywood tests usually use much rougher veneers, and this roughness leads to variable adhesive distribution, curing temperatures, and mechanical forces. Therefore, we tested some of our better formulations from the ABES results for bonding plywood using ASTM D 906 [18]. The results in Table 4 indicate that the use of pMDI plus PAE was not much better than pMDI as the sole cross-linker, but better than no crosslinker, since neither flour has any wet strength without a cross-linker. However, the PAE has a more significant impact on the dry and wet strengths when added to the canola as compared to the soy. This is perhaps because compared to soy, canola has more carbohydrate content, which is known to react with PAE. Also noted was that the mode of failure had changed from strictly cohesive in the ABES testing to more of an adhesion/cohesion failure. Thus, although the ABES provides a more controlled method to examine certain characteristics of adhesive performance, it does not predict adhesive performance in standard plywood tests.

### 3.3. ATR-FTIR Spectroscopy of pMDI Plus Soy or Canola Flour

Soy or canola flour was reacted with 50% pMDI and the dry mixture was analyzed by ATR-FTIR to understand the reaction with the isocyanates. As shown in Figure 3, the pMDI had a very large absorption band at 2250 cm^−1^, which almost completely disappeared from the soy flour plus pMDI mixture after 4 days. The pMDI band in the canola plus pMDI mixture was reduced by 75% after 4 days. The 2250 cm^−1^ absorption band is attributed to the N=C=O stretching vibrations [21].

There were many other bands from the pMDI that also reduced with time, but no new bands were identified that could be from the flours reacting with the pMDI. The reduction in the 2250 cm^−1^ band may be the flours reacting with the pMDI or it may be the pMDI self-polymerizing. When the soy or canola plus 50% pMDI mixture was dispersed in water at 25% solids (based on the flour) after 5 days as a dry mixture, the soy and canola adhesives retained most of their wet strengths in the ABES test (as shown in Figure 1). This suggests that the pMDI is reacting more with the flours as opposed to self-polymerizing and becoming less available for wood bonding.

## 4. Conclusions

The ABES data in this paper supports our prior observation that adding the pMDI to soy or canola flour provides a relatively good adhesive that can bond wood with greater water resistance than by adding the pMDI to an aqueous soy or canola flour dispersion. The data also shows that this pMDI–flour adhesive keeps its improved properties for about a week, but with a steady decline in wet bond shear strength.

Both the pMDI and PAE cross-linkers added to the flours increased the wet ABES bond shear strength using hard maple with an increase in the cross-linker amount before plateauing. However, the combination of the two cross-linkers added to the flours provides higher wet strength than is obtainable by either one separately.

## 5. Patent

Linda Lorenz and Charles Frihart have a patent pending to the U.S. government on the work reported in this manuscript (application number 17197956 was filed on 15 June 2021).

## Figures and Tables

**Figure 1 polymers-14-01272-f001:**
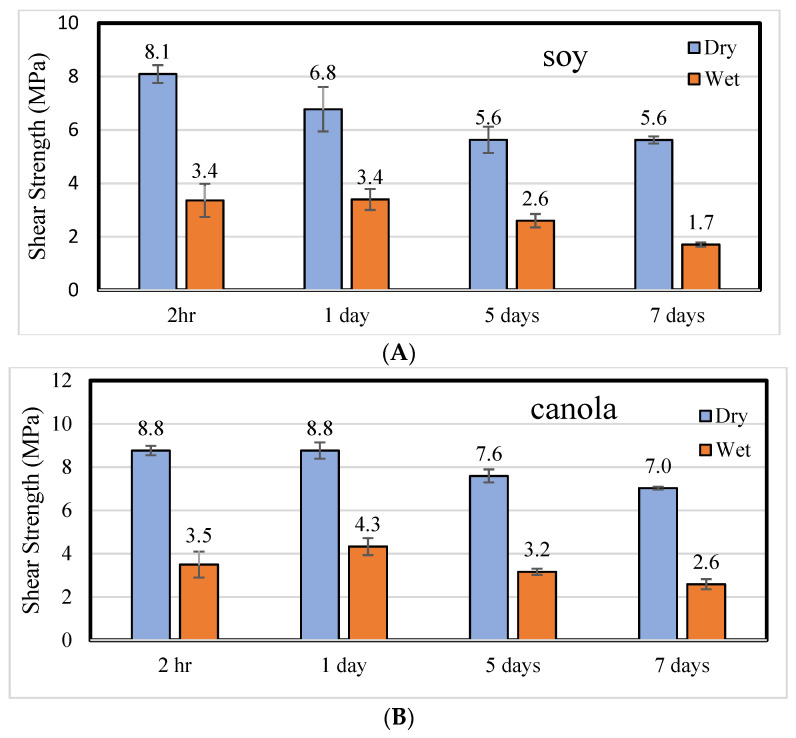
ABES shear strength (MPa), dry and wet, of the soy (**A**) and canola (**B**) adhesives (25% flour solids) with 50% pMDI (50 g pMDI/100 g flour) added to the dry flour and water added later.

**Figure 2 polymers-14-01272-f002:**
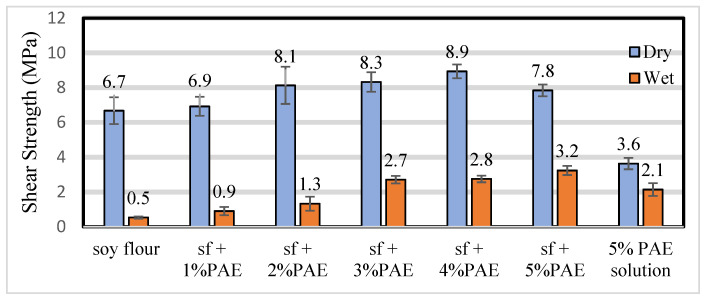
ABES shear strength (MPa), dry and wet, of the soy adhesives (sf, 25% solids) with very low amounts of PAE and a 5% PAE solution with no soy flour.

**Figure 3 polymers-14-01272-f003:**
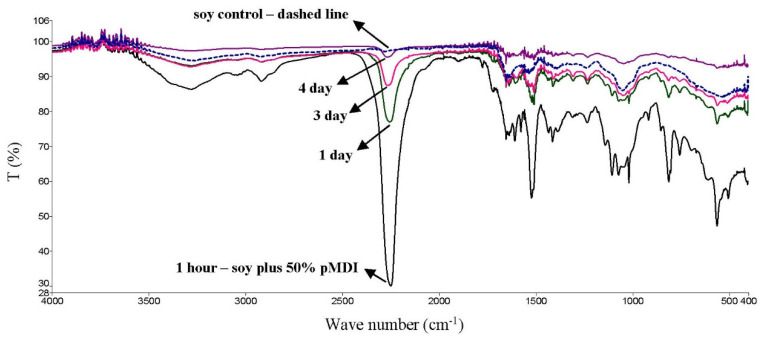
Soy flour reacted with 50% pMDI analyzed with FTIR after 1 h and 1, 3, and 4 days. The soy flour control with no pMDI is the dashed line.

**Table 1 polymers-14-01272-t001:** ABES shear strength (MPa) of canola and soy flour adhesives tested dry (ambient conditions) and wet (water soaked) compared to CSPI and our preferred minimum value for further testing.

Adhesives	Dry (MPa)	stdv	Wet	stdv
soy flour	5.3	0.8	0.3	0.09
canola flour	4.4	0.5	0.2	0.002
CSPI	8.2	1.6	3.2	0.3
minimum			3.0	

**Table 2 polymers-14-01272-t002:** ABES shear strength (MPa) of 25% soy or canola flour adhesives with increasing amounts of pMDI (in g/100 g of soy or canola flour) under dry and wet conditions.

Soy Flour 100 g	Dry	stdv	Wet	stdv	Canola Flour 100 g	Dry	stdv	Wet	stdv
+ pMDI 2.5 g	6.7	0.3	1.3	0.2	+ pMDI 2.5 g	5.6	0.4	1.4	0.2
+ pMDI 5 g	8.2	0.6	2.5	0.3	+ pMDI 5 g	6.6	0.2	1.8	0.0
+ pMDI 25 g	7.4	0.5	2.3	0.1	+ pMDI 25 g	7.5	0.1	2.4	0.1
+ pMDI 50 g ^1^	8.3	0.2	3.9	0.3	+ pMDI 50 g	8.8	0.2	3.5	0.6

^1^ This is equivalent to 50% pMDI of soy weight.

**Table 3 polymers-14-01272-t003:** ABES shear strength (MPa), dry and wet, of soy or canola adhesives (25% flour solids) with 50 g pMDI/100 g flour or with 50 g pMDI/100 g flour plus 5 g PAE solids/100 g flour.

Adhesives	Dry	stdv	Wet	stdv
soy flour + 50 g pMDI	8.3	0.2	3.9	0.3
soy flour + 50 g pMDI + 5 g PAE	7.9	0.5	4.9	1.2
canola + 50 g pMDI	8.8	0.2	3.5	0.6
canola + 50 g pMDI + 5 g PAE	9.8	0.2	5.6	0.3

**Table 4 polymers-14-01272-t004:** D 906 plywood test results of soy and canola flour (25% flour solids) dry and wet strength (MPa).

D 906 Samples	Dry (MPa)		Wet (MPa)	
	ave	stdv	ave	stdv
soy flour + 50 g pMDI + 5 g PAE	0.97	0.08	0.48	0.09
soy flour + 50 g pMDI	0.87	0.07	0.44	0.06
canola flour + 50 g pMDI + 5 g PAE	1.12	0.06	0.63	0.06
canola flour + 50 g pMDI	0.91	0.10	0.44	0.04

## Data Availability

There are no publicly archived datasets analyzed or generated during the study.

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
