# Peer review of "Improved Wood-Bond Strengths Using Soy and Canola Flours with pMDI and PAE"

_polymers, 2022, doi:10.3390/polym14071272_

Round 1
Reviewer 1 Report
- Authors have published many papers about improving wood bond strengths using soy and canola flours. The novelty and innovation need to explain.
- Bio-based adhesives are getting more attention since they are environmentally friendly green products without any adverse effects on the environment. There have been numerous studies of this in the past. Authors should summarize some of the research in this area. Some past works for your reference.
(https://www.mdpi.com/2073-4360/9/7/261
https://www.sciencedirect.com/science/article/pii/S0926669020308153
https://www.sciencedirect.com/science/article/pii/S0926669020311043
https://www.sciencedirect.com/science/article/pii/S0143749621001901
https://www.mdpi.com/2073-4360/13/7/1088
https://www.mdpi.com/2073-4360/9/4/132)
- ATR-FTIR mode is unsuitable for the FTIR test of Plywood. Besides, no references are added in the 3.3 part.
- The total references need to more.
Reviewer 2 Report
Dear Authors,
This is a well-planned and written article. I only have a few remarks that I present in synthetic form.
Materials – line 87-89
The trade and Latin names of the wood species are standardized (e.g. EN 13556:2003 Round and sawn timber – Nomenclature of timbers used in Europe).
The full names of the tested wood should be given: rock maple (Acer saccharum Marash.) white poplar (Populus alba L.) The description of the type of veneers is not complete. Are they circumferential or flat-cut veneers? The surface parameters of the veneers affect the gluing process and, as a result, the strength of the joints.
The recording of units should be standardized throughout the manuscript e.g. 100 g m-2 in line 109 and 215 g/m2 in line 109
Results and discussion - Figures 1 and 2
The way of filling bar surfaces with patterns in the charts is not accurate. I suggest using uniformly white filling for dry samples and uniformly light gray filling for wet samples. It will be much clearer and the marked deviations will be clearly visible.
Editing errors:
There is no space (leading) between the tables and adjacent text.
Some tables are unevenly positioned (tables 2 and 4).
The font size in the text is also not standardized and sometimes it changes randomly (line 204, lines 276-287, line 304).
Conclusions
The conclusions should be supplemented with the names of the tested wood species, as they refer to the gluing of specific veneers (veneers from hardwood with a low content of unstructured compounds). Tests on a different wood would probably give a different relationship.
Yours sincerely
Reviewer
Author Response
The names and descriptions of the veneers has been added.
The units have been standardized.
The figures have been changed to be more readable.
The editing errors have been corrected, even though most of the editing errors were not in the submitted
ms, especially the positioning of tables 2 and 4 and the lack of lines between them and the text.
The wood species were added to the conclusions.
Round 2
Reviewer 1 Report
Accept in present form
Reviewer 2 Report
Thank you for introducing additions and corrections to the manuscript in line with my suggestions in the first review.